# Relational Invariant Learning for Robust Solvation Free Energy Prediction

**Yeyun Chen** [1] [2]

## Abstract

Predicting the solvation free energy of molecules using graph neural networks holds significant potential for advancing drug discovery and the design of novel materials. While previous methods have demonstrated success on independent and identically distributed (IID) datasets, their performance in out-of-distribution (OOD) scenarios remains largely unexplored. We propose a novel Relational Invariant Learning framework (RILOOD) to enhance OOD generalization in solvation free energy prediction. RILOOD comprises three key components: (i) a mixup-based conditional modeling module that integrates diverse environments, (ii) a novel multi-granularity refinement strategy that extends beyond core substructures to enable context-aware representation learning for capturing multilevel interactions, and (iii) an invariant learning mechanism that identifies robust patterns generalizable to unseen environments. Extensive experiments demonstrate that RILOOD significantly outperforms state-of-the-art methods across various distribution shifts, highlighting its effectiveness in improving solvation free energy prediction under diverse conditions.

## 1. Introduction

Predicting the solvation free energy of molecules is crucial, as most chemical and pharmaceutical processes occur in solution, making it highly significant for downstream industries (Chung et al., 2022; Varghese & Mushrif, 2019). This task, often referred to as Solute-Solvent Interaction in Molecular Relational Learning (MRL) (Lim & Jung, 2019; Subramanian et al., 2020; Panwar et al., 2021; Low et al., 2022; Zhang et al., 2022; Lee et al., 2023a;b), focuses on understanding and modeling the interactions between solutes and solvents, conceptualizing these interactions as

solvation properties of molecules. More importantly, it extends traditional molecular property prediction frameworks by explicitly incorporating solvent molecules as input features, thereby improving prediction accuracy and enhancing chemical interpretability.

Despite significant advancements in MRL, most existing methods operate under the assumption that training and test data are independent and identically distributed (IID). However, real-world molecular systems exhibit diverse characteristics and uneven data distributions across solvents, making this assumption unrealistic in practical applications. Out-of-Distribution (OOD) scenarios arise when test data differs substantially from training data, as illustrated by the example in Fig. 1. This work focuses on exploring the OOD generalization of molecular solvation properties across different environments within the MRL.

To address distribution shifts, several approaches have been proposed, including invariant learning (Wu et al., 2022a), feature disentanglement (Liu et al., 2021), and data augmentation (Sui et al., 2024; Jia et al., 2024). Among these, invariant learning for OOD generalization (Krueger et al., 2021) has garnered significant attention due to its ability to extract robust features that remain stable across different environments, even under distribution shifts. In molecular modeling, molecular invariant learning is commonly employed to address distribution shifts by identifying core substructures that exhibit strong correlations with molecular properties. However, existing methods encounter notable limitations. For example, Lee et al. (2023a) leverage privileged substructures as causal correlations in MRL. However, they do not account for the solvent-dependent nature of solute properties and the complex coupling effects that govern molecular behavior (Cramer & Truhlar, 2008). Similarly, Lee et al. (2023b) apply back-door adjustment to mitigate spurious correlations but fails to account for solvent effects, thereby neglecting intricate solute-solvent interactions that are essential for accurately characterizing solute properties. As a result, these approaches lead to an incomplete understanding of solute behavior, increased susceptibility to spurious correlations, and poor generalization to unseen environments.

Although atomic interactions have been extensively modeled and have shown success in MRL, a precise under-

---

[1]Institute of Artificial Intelligence, Xiamen University, China [2]Shanghai Innovation Institute, China. Correspondence to: Yeyun Chen <yeyunchen2022@stu.xmu.edu.cn>.

*Proceedings of the $42^{nd}$ International Conference on Machine Learning*, Vancouver, Canada. PMLR 267, 2025. Copyright 2025 by the author(s).

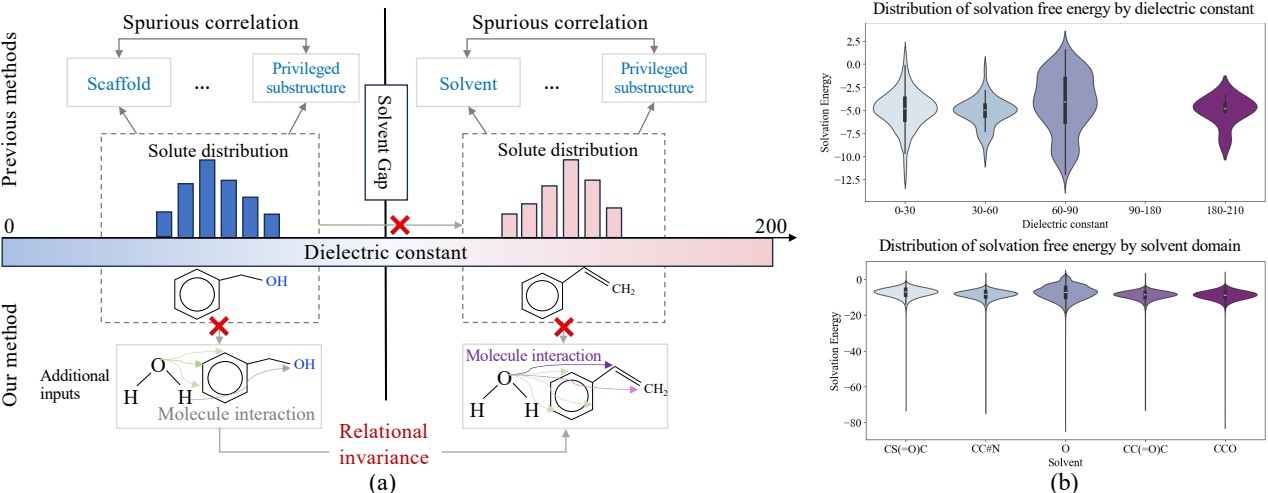

*Figure 1.* Illustration of solvation free energy distribution across different solvents. (a) illustrates how the solvation properties of solute molecules vary with changes in the solvent environment. The horizontal axis represents the dielectric constant of solvent, and the dotted boxes indicate the distribution of solute properties in two solvents: benzyl alcohol (blue, left) and styrene (pink, right). Traditional methods predict solute properties by identifying core substructures, but core substructures may change in different solvents. (b) presents the distribution of solvation energy across different solvent dielectric constants (top) and various solvents (bottom), highlighting the significant influence of the solvent on solvation energy. Highly polar solvents, or those with intermediate dielectric constants, tend to exhibit stronger interactions with solutes, leading to a broader range of solvation energies. Relational invariance refers to molecular interactions such as hydrogen bonding (purple) and van der Waals forces (gray). The statistical data originates from the MNSolv dataset (Marenich et al., 2012) (top) and the QM9Solv dataset (Ward et al., 2021) (bottom).

standing of the solute-solvent interaction remains elusive. Achieving both accuracy and explainability presents significant challenges, particularly in the following areas: (1) the need to accurately model solute–solvent interactions across chemically diverse solvent environments. (2) the inherent complexity of multilevel molecular interactions, which hinders the extraction of invariant features and the construction of robust, generalizable representations. To address these issues, it is essential to develop models that can accurately represent multilevel molecular interactions, effectively capturing the complex one-to-many relationships between solutes and properties.

Based on the aforementioned analysis, in this work, we propose a novel Relational Invariant Learning framework for Out-of-Distribution Generalization (RILOOD) in MRL. Unlike traditional methods, our framework explicitly captures invariant relationships in molecular pairs and achieves a more generalized representation of solute-solvent interactions. Specifically, we first employ a Graph Neural Network (GNN) to encode molecular structures, followed by a cross-attention module to map atom-level interactions. We then incorporate mixup-enhanced Conditional Variational Modeling to facilitate cross-environment invariance, leveraging a multi-granularity context-aware interaction mechanism and environment diversity inference. This enables learning of interaction invariance (Xie et al., 2024), allowing the discov-

ery of fundamental molecular relationships in a chemically interpretable latent space. Our main contributions can be summarized as follows:

- We formally formulate the out-of-distribution (OOD) generalization problem in Molecular Relational Learning (MRL), establishing a rigorous foundation for studying model robustness across diverse chemical environments.

- We propose RILOOD, a relational invariant learning framework for solvation free energy prediction, featuring three key components: a mixup-based conditional modeling module, a multi-granularity refinement strategy, and an invariant learning mechanism.

- We conduct extensive experiments across multiple distribution shifts, demonstrating that RILOOD consistently outperforms state-of-the-art methods, significantly advancing OOD generalization in molecular property prediction.

## 2. Related Works

### 2.1. Molecular Relational Learning

Molecular Relational Learning (Lim & Jung, 2019; Pathak et al., 2020; Subramanian et al., 2020; Lee et al., 2023a;b), which aims to study the relationship between molecules,

can be divided into molecular interaction prediction and Drug-Drug Interaction prediction. Molecular interaction prediction, i.e., solvent-based molecular property prediction, includes solvent free energy prediction, solubility prediction, chromophore absorption prediction, and so on. Unlike traditional molecular property prediction, the model need predict the properties exhibited by the same molecule exposed to multisolvent. Recent works (Ramani & Karmakar, 2024; Du et al., 2024) leverage the merged graph to encode atomic interaction and further improve interpretability and reducing redundancy.

### 2.2. Out of Distribution Generalization

Generalizing well-trained models to unseen environments with different data distributions remains a key challenge in machine learning. To address OOD generalization, three main approaches are typically employed: invariant learning (Li et al., 2022), causal inference (Dawid, 2000), and disentangled learning (Mo et al., 2023). Invariant learning aims to extract stable features across distribution shifts, but ZIN (Lin et al., 2022) argues that identifying invariance in Euclidean data is impossible without environment labels, proposing auxiliary information as a solution. Causal inference approaches utilize Structural Causal Models (SCM) (Chen et al., 2022; Lu et al., 2021) and Independent Causal Mechanisms (ICM) (Peters et al., 2017; Gui et al., 2024) to filter spurious correlations and enhance robust feature discovery. Disentangled learning separates features into invariant factors, which generalize across distributions, and spurious factors, which exhibit unstable correlations. While effective, it relies on strong prior assumptions and carefully curated datasets. These diverse strategies collectively tackle OOD generalization by distinguishing stable predictive patterns from environment-dependent variations, yet significant challenges remain in accurately identifying and effectively leveraging invariant features.

### 2.3. Invariant Learning in Molecular Relational Learning

Research on invariant learning in molecular representation learning remains sparse. One approach identifies core substructures using the graph information bottleneck to extract minimal task-relevant information (Lee et al., 2023a). Another method leverages causal intervention to learn causal substructures and mitigate distribution shifts (Lee et al., 2023b). In OOD settings, generalization is typically evaluated by partitioning datasets into scenarios like "unseen solvent" or "unseen domain", where test sets exhibit specific biases. However, many studies remain confined to intra-domain frameworks, failing to capture real-world complexities. Despite successes in graph-based invariant learning (Wu et al., 2022a; Yang et al., 2022; Li et al., 2022), two key challenges persist: (1) Environmental labels for graphs

are difficult to obtain, often relying on handcrafted rules that provide insufficient causal structure. (2) Invariant patterns and spurious correlations are entangled with shortcut features, complicating the identification of stable representations. Addressing these challenges is crucial for improving the robustness and generalization of MRL models across diverse molecular environments.

## 3. Preliminaries

We define the uppercase letters (e.g., $\mathcal{G}$) as random variables, and the blackboard typefaces (e.g., $\mathbb{G}$) denote the sample spaces. Let $\mathcal{G} = (\mathcal{V}, \mathcal{E}) \in \mathbb{G}$ denote a graph, where $\mathcal{V} = \{v_1, v_2, ..., v_n\}$ is the set of nodes and $\mathcal{E} \in \mathcal{V} \times \mathcal{V}$ is the set of edges.

### 3.1. Molecular Relational Learning.

The goal of MRL task is to predict the target label $\mathcal{Y}$ given the associated input molecular pairs $(\mathcal{G}_1, \mathcal{G}_2)$. It can be formulated as modeling the conditional distribution $p(\mathcal{Y}|\mathcal{G}_1, \mathcal{G}_2)$.

**Notations.** Given a dataset $\mathcal{D} = \{((\mathcal{G}_1^i, \mathcal{G}_2^i), \mathcal{Y}^i)\}_{i=1}^N$, where $\mathcal{G}_1 \in \mathbb{G}_1$ is solute molecule, and $\mathcal{G}_2 \in \mathbb{G}_2$ is solvent molecule, each molecular pair is associated with a target label $\mathcal{Y}$. $N$ is the total number of samples. The objective is to train a model to predict $\mathcal{Y}$ based on the input $(\mathcal{G}_1, \mathcal{G}_2)$. The model should effectively learn the relationships between the input features and the target variable, leveraging the information from both $\mathcal{G}_1$ and $\mathcal{G}_2$ to accurately predict $\mathcal{Y}$. The model's performance will be evaluated based on the RMSE of the predicted output $\hat{\mathcal{Y}}$ in comparison to the ground truth labels $\mathcal{Y}$.

**Molecular Representation.** We implement our method based on (Pathak et al., 2020), which is a message passing architecture devised for the solute and solvent molecule interaction. Given a pair of molecules $\mathcal{G}_1 = (\mathcal{V}_1, \mathcal{E}_1)$ and $\mathcal{G}_2 = (\mathcal{V}_2, \mathcal{E}_2)$. We first obtain the node representation of each molecule as follows: $h_1 = \text{GNN}(\mathcal{V}_1, \mathcal{E}_1), h_2 = \text{GNN}(\mathcal{V}_2, \mathcal{E}_2)$. To capture inter-molecular interactions at the atomic level, the interaction map is constructed as following: $I = h_1 \cdot h_2^T$, where $\cdot$ is matrix multiplication, $I \in \mathbb{R}^{N_1 \times N_2}$. Here, $N_1$ and $N_2$ denote the number of atoms in molecule $\mathcal{G}_1$ and $\mathcal{G}_2$, respectively. We obtained a representation $\tilde{h}_1 \in \mathbb{R}^{N_1 \times D}$ of the solvent's interaction on the solute and a representation $\tilde{h}_2 \in \mathbb{R}^{N_2 \times D}$ of the solute's interaction on the solvent through a shared interaction map according to the following equations: $\tilde{h}_1 = I \cdot h_2, \tilde{h}_2 = I^T \cdot h_1$. $H_1$ is generated by concatenating two representations $\tilde{h}_1$ and $h_1$, i.e., $H_1 = concat[h_1, \tilde{h}_1]$. The overall graph representation is obtained using a readout layer $R_{solute}(H_1)$, which set the READOUT function as Set2Set (Vinyals et al., 2015).

## 3.2. OOD Generalization.

In this work, we mainly focus on OOD generalization in graph-level prediction tasks. Our aim is to train the model with limited labels to infer the domain distribution from unseen data in $\mathcal{D}_{te}$.

**Problem formulation.** Given a molecular pairs dataset, $\mathcal{D} = \{((\mathcal{G}_1^i, \mathcal{G}_2^i), \mathcal{Y}^i)_{i=1}^{N^{tr+te}}\}$ collect from multiple environments $\mathcal{E}$, which were considered as drawn independently from an identical distribution $p_e$, i.e., $\mathcal{D}_{ID} = \{(\mathcal{G}_1, \mathcal{G}_2) \in \mathcal{D} \mid \mathcal{G}_1 \in \mathbb{G}_{ID} \bigwedge \mathcal{G}_2 \in \mathbb{G}_{ID}\}$. The training and test datasets are denoted as $\mathcal{D}_{tr} = \{((\mathcal{G}_1^i, \mathcal{G}_2^i), \mathcal{Y}^i)\}_{i=1}^{N^{tr}}$ and $\mathcal{D}_{te} = \{((\mathcal{G}_1^i, \mathcal{G}_2^i), \mathcal{Y}^i)\}_{i=1}^{N^{te}}$. Our goal is to find an optimal predictor $\Phi: (\mathbb{G}_1, \mathbb{G}_2) \to \mathbb{Y}$ that performs well on all environments. Formally, the learning objectives can be formulated as:

$$\min_{\Phi} \max_{e \in \mathcal{E}} \mathbb{E}_{((\mathcal{G}_1^i, \mathcal{G}_2^i), \mathcal{Y}^i) \sim p((\mathbb{G}_1, \mathbb{G}_2), \mathbb{Y}|e)} \left[ \ell \left( \Phi \left( \mathcal{G}_1^i, \mathcal{G}_2^i \right), \mathcal{Y}^i \right) \right] \tag{1}$$

**Definition 3.1.** (Data generation process) The OOD distribution can be sampled according to $\mathcal{D}_{OOD} = \{(\mathcal{G}_1, \mathcal{G}_2) \in \mathcal{D} \mid (\mathcal{G}_1 \in \mathbb{G}_{OOD} \bigwedge \mathcal{G}_2 \in \mathbb{G}_{OOD}) \bigvee (\mathcal{G}_1 \in \mathbb{G}_{OOD} \bigwedge \mathcal{G}_2 \in \mathbb{G}_{ID}) \bigvee (\mathcal{G}_1 \in \mathbb{G}_{ID} \bigwedge \mathcal{G}_2 \in \mathbb{G}_{OOD})\}$. The data generation process is as follows: Let $\mathcal{E}$ denote all possible environments, $supp(N_{tr}) \subset supp(\mathcal{E})$, sampled train data from $p((\mathcal{G}_1, \mathcal{G}_2), \mathcal{Y})$. Out of Distribution indicate that $p_e((\mathcal{G}_1, \mathcal{G}_2), \mathcal{Y}) \neq p_e'((\mathcal{G}_1, \mathcal{G}_2), \mathcal{Y})$, i.e., $\mathcal{D}_{\text{train}} = \{((\mathcal{G}_1^i, \mathcal{G}_2^i), \mathcal{Y}^i)_{i=1}^{N^{tr}} \mid e \subset supp(N_{tr})\}$, $\mathcal{D}_{\text{test}} = \{((\mathcal{G}_1^i, \mathcal{G}_2^i), \mathcal{Y}^i)_{i=1}^{N^{te}} \mid e' \in supp(\mathcal{E}) \backslash supp(N_{tr})\}$.

Details of the OOD dataset splitting are provided in the Appendix B.1.

## 4. Methodology

In this section, we introduce a **R**elational **I**nvariant **L**earning framework designed to address **O**ut-**o**f-**D**istribution generalization (**RILOOD**) in solvation free energy prediction. An overview of the proposed method is provided in Fig. 2. We detail the motivations and technical aspects of the three key components in RILOOD: **Mixup-enhanced Conditional Variational Modeling** (Section 4.2), **Multi-granularity Context-Aware Refinement** (Section 4.3), and **Invariant Relational Learning Mechanism** (Section 4.4).

### 4.1. The Overall Framework.

In molecular relation learning, substructure identification methods are widely employed. However, these approaches often fail to account for the variability in a solute's behavior across different solvents, as solute-solvent interactions can differ significantly.

To overcome this limitation, invariant learning aims to identify stable features or patterns that remain consistent across

diverse environments. This approach reduces prediction errors and minimizes dependence on environmental variations. A predictor performing well across multiple, varied environments is more likely to generalize robustly to unseen distributions. Our primary objective is to develop a model that is robust to domain shifts, ensuring the mapping from molecule pairs to labels remains stable irrespective of environmental changes.

**Assumption 4.1.** Given a molecular pair $(\mathcal{G}_1, \mathcal{G}_2)$, each pair is associated with $R$ surrounding environments. We assume the existence of invariant interaction patterns that facilitate generalizable OOD predictions across all environments. The optimal predictor $\Phi(\cdot)$ should satisfy the properties of *Invariance* and *Sufficiency*, as detailed in Appendix A.3.

Specifically, we further decompose $\Phi(\cdot)$ into two key components as $\Phi(\cdot) = g \circ f(\mathcal{G}_1, \mathcal{G}_2)$: (a) A Conditional Variational Autoencoder (CVAE) $f$, which models the prior distribution of the solute representation $H_1 \sim p_e(z|e)$ across different environments. Here, $z$ is a low-dimensional, continuous representation of the solute in the latent space. (b) A multi-granularity context-aware learner $g$, which refines relational features by mapping $(H_1, H_2)$ to a context-aware representation $H_c$, i.e., $g: (H_1, H_2) \to H_c$.

Building on Eq. 1, we reformulate the OOD generalization problem for molecular pairs as:

$$\min_{g} \max_{e \in \mathcal{E}} \mathbb{E}_{(\mathcal{G}_1^i, \mathcal{G}_2^i, \mathcal{Y}^i) \sim p(\mathbb{G}_1, \mathbb{G}_2, \mathbb{Y}|e)} \left[ \ell \left( g \circ f \left( \mathcal{G}_1^i, \mathcal{G}_2^i \right), \mathcal{Y}^i \right) \right] \tag{2}$$

where $e$ denotes the support environments, and $\ell(\cdot, \cdot)$ represents the loss function.

### 4.2. Mixup-enhanced Conditional Variational Modeling

Empirically, acquiring explicit environmental labels for solute–solvent pairs is often impractical, presenting a major obstacle to learning solute representations that generalize across diverse solvent conditions. This challenge is further compounded by limited data coverage, which may fail to capture distributional shifts induced by variations in solvation environments. Fundamentally, the difficulty in obtaining environmental labels arises from the fact that solvation energy is governed by complex, multi-scale interactions between solutes and solvents, including non-covalent intermolecular forces and functional group–specific dependencies. In the absence of explicit labels, these underlying physicochemical mechanisms are difficult to accurately model or disentangle. To address this challenge, ZIN (Lin et al., 2022) introduces a method for inferring latent environmental partitions using auxiliary information. Inspired by this approach, we propose conditioning on auxiliary information to implicitly capture solute representations across the environment.

**Mixup Enhanced.** To further improve generalization, we

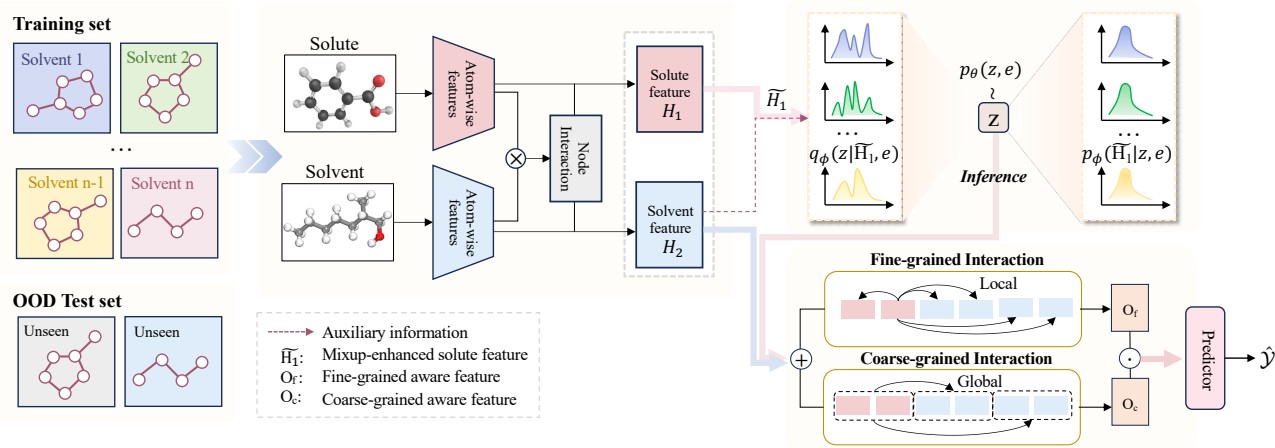

**Figure 2.** Our proposed RILOOD framework adopts an architecture that integrates mixup-enhanced conditional variational modeling and multi-granularity context interaction mechanisms to provide an efficient way to adapt across different scenarios.

adopt mixup, which generates interpolated representations to enhance robustness across environments. We propose Mixup-enhanced CVAE (MCVAE), a module designed to model molecular distributions using paired solvent information and infer the latent distribution $q_\phi(z|\tilde{H}_1, e)$ across diverse environments. Furthermore, we introduce uncertainty constraints to regularize the latent space, thereby improving stability and generalization across varying solvent conditions.

We assume that solvents belong to $R$ discrete categories, denoted as $\mathcal{E} = \{e^r\}_{r=1}^R$, where each solvent type $e^r$ is represented as a $R$-dimensional one-hot vector $e^r \in \{0,1\}^R$, with its $r$-th dimension set to 1. To enhance generalization, we employ mixup augmentation to interpolate between environmental conditions, allowing the model to learn continuous latent representations rather than relying solely on discrete labels. This encourages better adaptation to unseen domains.

Given the molecular representations $H_1$ and $H_2$ for molecules $\mathcal{G}_1$ and $\mathcal{G}_2$, respectively, we construct enhanced samples using the following mixup formulation:

$$\tilde{H}_1 = \lambda \cdot H_1 + (1-\lambda) \cdot H_2, \quad e = \lambda \cdot e_1 + (1-\lambda) \cdot e_2 \quad (3)$$

where $\tilde{H}_1$ denotes the mixed molecular representation and $e$ signifies the interpolated solvent condition. The mixing coefficient $\lambda$, is sampled from a Beta distribution, specifically $\lambda \sim \text{Beta}(\alpha, \alpha)$, to ensure a smooth interpolation between different environments. This modeling strategy can be readily extended to other contexts, such as scaffold-based modeling. The derivations for the upper and lower bounds of this mixed representation are detailed in Appendix A.1.

For the regression task, we incorporate an uncertainty constraint to mitigate noise introduced by the mixup technique.

Here, $\sigma(\cdot)^2$ is the uncertainty variance, and we constrain it. Consequently, the regression loss can be reformulated as follows:

$$\mathcal{L}_{reg} = \frac{1}{N} \sum_{i=1}^N \left[ \frac{1}{\sigma(H_1^i)^2} \|y_i - \Phi(\mathcal{G}_1^i, \mathcal{G}_2^i)\|^2 + \log \sigma(H_1^i)^2 \right] \quad (4)$$

**Variational Inference for Mixup Representations.** To model the conditional log-likelihood $\log p(\tilde{H}_1 \mid e)$, we introduce variational inference, reformulating the objective as a variational lower bound by approximating the posterior distribution $q(z \mid \tilde{H}_1, e)$:

$$\max_{\theta, \phi} \mathbb{E}_{\tilde{H}_1 \sim D} \left[ \mathbb{E}_{q_\phi(z|\tilde{H}_1, e)} \left[ \log p_\theta(\tilde{H}_1 \mid z, e) \right] \right],$$
$$\text{s.t.} \quad D_{KL}\left( q_\phi(z \mid \tilde{H}_1, e) \parallel p_\theta(z \mid \tilde{H}_1) \right) < \delta \quad (5)$$

where $\delta$ is a threshold, and the $KL$ divergence constraint ensures that the approximate posterior remains close to the prior, preventing latent space collapse and enhancing generalization across diverse solvent conditions.

In practice, as the number of solvent categories grows, independent modeling becomes computationally prohibitive. To enhance scalability, a learnable function could be employed to assign soft environment labels to solvents, rather than modeling each solvent separately. However, given the limited solvent diversity in our dataset (e.g., only five solvents in QM9Solv dataset), we refrain from adopting this approach to ensure a fair comparison.

**Environment Inference via MCVAE Optimization.** By optimizing MCVAE, we aim to infer the solute distribution within a latent environment, offering a novel strategy for learning environment-aware molecular representations.

Specifically, we minimize the divergence between the approximate posterior distribution $q_\phi(z \mid \tilde{H}_1, e)$ of the latent variable $z$ and the true posterior probability $p_\phi(\tilde{H}_1 \mid e, z)$ for a specific environment $e$. Leveraging the rich prior knowledge embedded in conditional encoders, our training objective comprises two main components: (1) Encourage the $R$-group latent distribution produced by the encoder to approximate a standard normal distribution as closely as possible. (2) Sample $z$ from the conditional distribution $q_\phi(z \mid e)$, and ensure that the reconstructed solute molecular features closely resemble the original features.

$$\mathcal{L}_{\text{MCVAE}}(\theta, \phi; \tilde{H}_1, e) = -KL\left(q_\phi(z \mid \tilde{H}_1, e) \| p_\theta(z \mid \tilde{H}_1)\right)$$
$$+ \frac{1}{N} \sum_{i=1}^{N} \left[\frac{1}{\sigma(\tilde{H}_1)^2} \|z - \tilde{H}_1\|^2 + \log \sigma(\tilde{H}_1)^2\right]$$
$$(6)$$

where $z = g_\phi\left(\tilde{H}_1, e, \epsilon\right)$, with $\epsilon \sim \mathcal{N}(0, \mathbf{I})$, and $\mathbf{I}$ is the identity matrix. Detailed proof is in Appendix A.2.

### 4.3. Multi-granularity Context-Aware Refinement.

Although the solvent type serves as a useful proxy for environmental context, directly relying on it can lead to predictive shortcuts. To address this, we propose a joint optimization framework that simultaneously learns: (i) an environment modeling function from auxiliary information, (ii) an interaction-aware feature extractor, and (iii) mutual information constraints. The goal is to encode a context-aware molecular representation that dynamically captures solute behavior under specific conditions while accurately modeling the context-dependent relevance of solute molecules.

Previous studies have predominantly focused on atomic-level interactions within molecules (Pathak et al., 2020). However, since solvents interact globally with solutes, a multi-granularity interaction strategy is better suited to capture their complex effects. Notably, solute-solvent interactions are non-covalent and are often neglected in conventional modeling approaches. To address this limitation, we employ a self-attention mechanism to dynamically identify intermolecular interactions, allowing representations to adapt more effectively across diverse environments.

**Multi-granularity Interactions with Context.** To effectively model solute–solvent interactions, we propose a Multi-granularity Context-Aware Refinement (MCAR) strategy, designed to capture hierarchical interaction patterns across multiple levels. The MCAR mechanism is implemented in two key steps: (1) Context learning: Both coarse-grained molecular-level contexts and fine-grained feature-level contexts are simultaneously captured to jointly learn comprehensive contextual information. (2) Pattern refinement: Invariant interaction patterns are refined through matrix multiplication between the coarse-grained and fine-grained feature representations.

We begin by constructing an initial embedding $E$ that aggregates relevant molecular features: $E = \text{concat}[z, H_2]$, where $z$ and $H_2$ represent specific molecular features. To model global interactions, we project $E$ into query (Q), key (K), and value (V) matrices using learnable linear transformations:

$$Q, K, V = EW_Q, \ EW_K, \ EW_V \qquad (7)$$

where $W_Q, W_K, W_V$ are trainable projection matrices. A scaled dot-product self-attention mechanism is then applied to these matrices to capture long-range dependencies across molecular entities:

$$O_c = Attention(Q, K, V)$$
$$= W_c \cdot \text{Softmax}\left(\frac{QK^T}{\sqrt{d_k}}\right) V \qquad (8)$$

where $W_c$ is learnable transformations that enhance attention expressiveness. In parallel, local intra-molecular interactions are captured via a non-linear transformation:

$$O_f = \text{PReLU}(W_L E + b_L) \qquad (9)$$

where $W_L$ and $b_L$ are learnable parameters for local feature transformation. The PReLU is activation function to enhance the expressiveness of the local representations. Finally, the context-aware representation is obtained by performing a Hadamard product between the global and local interaction features:

$$H_c = O_c \circ O_f \qquad (10)$$

where $\circ$ denotes the Hadamard product, enabling the integration of multi-scale contextual information into a unified and expressive feature representation.

To improve feature extraction, we maximize mutual information to retain essential features while reducing redundancy and noise. Specifically, we optimize the mutual information between the solute representation $H_1$ and the context-aware feature $H_c$. The solute feature $H_1$, may be influenced by spurious correlations that hinder generalization. In contrast, the context-aware feature $H_c$ captures invariant and meaningful correlations, leading to more robust representations across diverse environments. We therefore denote $H_c$ as $\widehat{H}_{\text{inv}}$ to emphasize its role in learning invariant features. To this end, we formulate the optimization objective as follows:

$$\max_{f_c, w} I\left(\widehat{H}_{inv}; \mathcal{Y}\right),$$
$$\text{s.t.} \widehat{H}_{inv} \in \underset{\widehat{H}_{inv}=w(H_1), |\widehat{H}_{inv}| \leq H_1}{\arg\max} I\left(\widehat{H}_{inv}; H_1 \mid \mathcal{Y}\right) \qquad (11)$$

Finally, contrastive learning provides a practical solution for

the approximation, the learning objective is defined as:

$$\mathcal{L}_{MI} =$$
$$-\frac{1}{M} \sum_{i=1}^{M} \log \frac{exp(sim(\hat{H}_{inv}^i, H_1^i))}{exp(sim(\hat{H}_{inv}^i, H_1^i)) + \sum_{j=1, j\neq i}^{M} exp(sim(\hat{H}_{inv}^i, H_1^j))} \quad (12)$$

### 4.4. Invariant Relational Learning Mechanism

**Optimization Objective.** Eq. 2 clarifies the training objective of OOD generalization. However, directly optimizing Eq. 2 is not impracticable. Instead, we formulate a joint optimization framework:

$$\mathcal{L} = \mathcal{L}_{\text{inv}} + \alpha \mathcal{L}_{\text{MCVAE}} + \beta \mathcal{L}_{\text{MI}} \quad (13)$$

where $\alpha$ and $\beta$ are weight hyperparameters for $\mathcal{L}_{\text{MCVAE}}$ and $\mathcal{L}_{\text{MI}}$, respectively. The term $\mathcal{L}_{\text{inv}}$ represents the prediction loss, measuring the discrepancy between the model's output and the ground truth. For regression tasks, Eq. 4 can be used instead of $\mathcal{L}_{\text{inv}}$.

**Proposition 4.2.** *Given the auxiliary environment e, our goal is to build a model $p_\theta(\tilde{H}_1 \mid e, z)$ that learns the feature $\tilde{H}_1 \in \mathbb{R}$ conditioned on e. Optimizing Eq. 6 ensures that z exhibits sufficient predictive power, thereby allowing the model to satisfy the Sufficient condition in Assumption 4.1. Furthermore, minimizing Eq. 13 encourages the model to satisfy the Invariance condition in Assumption 4.1.*

## 5. Experiments

In this section, we conduct extensive experiments to answer the research questions:

- **RQ1:** How to evaluate the effectiveness of the model in OOD scenarios?

- **RQ2:** How effective is RILOOD in discovering invariant features and improving generalization?

### 5.1. Experimental Settings

**Datasets.** We use six datasets to evaluate our method. Specifically, the Minnesota Solvation Database (MN-Solv) (Marenich et al., 2012), QM9Solv (Ward et al., 2021), CompSolv (Moine et al., 2017), CombiSolv (Vermeire & Green, 2021), MolMerger (Ramani & Karmakar, 2024), and Abraham (Grubbs et al., 2010). The detailed statistics and descriptions are given in Appendix B. More experiments are provided in Appendix C.2.

**Baselines.** For a comprehensive comparison, we adopt two types of baselines: (1) There is no interaction layer between molecular encoders. We use three commonly used GNN models, including GIN (Xu et al., 2018), GCN (Kipf &

Welling, 2016), GAT (Veličković et al., 2017), to obtain molecular embedding through concatenation and then enter the prediction layer; (2) there is an interaction layer between molecular encoders, including ERM (Vapnik, 2013), Group-DRO (Sagawa et al., 2019), MixUp (Zhang et al., 2017a), MolMerger (Ramani & Karmakar, 2024), CIGIN (Pathak et al., 2020), CGIB (Lee et al., 2023a), CMRL (Lee et al., 2023b).

**Metrics.** We choose widely-used metrics in previous works, the performance of the molecular interaction prediction task is evaluated in terms of RMSE (Pathak et al., 2020). Lower error indicate better prediction performance. AUROC (Lee et al., 2023b) for DDI prediction.

### 5.2. Main Results (RQ1)

**Real-world Dataset.** To assess the generalization performance of our method, we conducted comprehensive experiments on three datasets, demonstrating the effectiveness of the proposed approach. To further explore distribution shifts across diverse environments, we evaluated model performance under two distinct settings: Solvent and Scaffold. The overall results, summarized in Tab. 1, lead to the following key observations:

Our method consistently outperforms baseline models, achieving superior results across all datasets. Traditional approaches exhibit limitations, as they primarily rely on substructure-based invariance, which often introduces spurious correlations in MRL. The notable improvement observed in RILOOD stems from its ability to capture multi-granularity interactions and identify invariant patterns, enabling the model to effectively adapt to domain shifts. Further discussions on extending this method to the I.I.D. setting are provided in Appendix C.2, with additional results in Tab. 4.

**Synthetic Dataset.** To evaluate model robustness under distribution shifts, we apply dataset-specific shift strategies, introducing spurious features to construct synthetic datasets. Following (Li et al., 2022; Wu et al., 2022b), spurious correlations are injected by controlling the variant distribution. Further details are provided in Appendix C.2. Specifically, we manually introduce spurious relationships of varying degrees between environment $e$ and the label $\mathcal{Y}$ in the training set, setting degree $d = \{0.25, 0.33, 0.5, 0.75\}$. The results, presented in Fig. 3 (a), indicate that as $d$ increases, performance generally improves due to a greater degree of distribution shift. However, our proposed method exhibits the highest stability, effectively mitigating the effects of spurious correlations.

**Generalization on Graph Classification.** To evaluate the applicability of our method to molecular pair data and classification tasks, we conducted experiments on the DDI dataset.

*Table 1.* Performance comparison with baselines on 3 out-of-distribution real-world datasets from MNSolv, CompSolv, QM9Solv in terms of RMSE. Different dataset splits by specific shift (solvent split and scaffold split), and details can be find in Appendix B.1. The best and the runner-up results are highlighted in bolded and underlined respectively.

| Interaction | Method | MNSolv↓ | | CompSolv↓ | | QM9Solv↓ | |
|---|---|---|---|---|---|---|---|
| | | Solvent | Scaffold | Solvent | Scaffold | Solvent | Scaffold |
| ✗ | GCN | $0.8921_{\pm0.024}$ | $\underline{1.2752}_{\pm0.022}$ | $0.7644_{\pm0.024}$ | $0.9598_{\pm0.018}$ | $0.9115_{\pm0.052}$ | $1.0319_{\pm0.046}$ |
| | GIN | $0.7723_{\pm0.032}$ | $1.3685_{\pm0.049}$ | $0.5927_{\pm0.013}$ | $0.6004_{\pm0.028}$ | $1.0928_{\pm0.017}$ | $1.0762_{\pm0.052}$ |
| | GAT | $0.7022_{\pm0.012}$ | $1.4566_{\pm0.022}$ | $0.5338_{\pm0.064}$ | $0.5526_{\pm0.027}$ | $1.0513_{\pm0.042}$ | $1.0415_{\pm0.012}$ |
| ✓ | ERM | $0.7503_{\pm0.026}$ | $1.3478_{\pm0.013}$ | $0.5360_{\pm0.002}$ | $0.6334_{\pm0.003}$ | $0.7471_{\pm0.053}$ | $\underline{0.7261}_{\pm0.005}$ |
| | GroupDRO | $0.7839_{\pm0.003}$ | $1.4322_{\pm0.031}$ | $0.5857_{\pm0.013}$ | $0.7459_{\pm0.012}$ | $0.8259_{\pm0.007}$ | $0.8503_{\pm0.021}$ |
| | MixUp | $\underline{0.7135}_{\pm0.011}$ | $1.3843_{\pm0.012}$ | $0.5772_{\pm0.026}$ | $0.5604_{\pm0.017}$ | $\underline{0.7227}_{\pm0.003}$ | $0.7490_{\pm0.002}$ |
| | MolMerger | $0.9276_{\pm0.081}$ | $1.6225_{\pm0.046}$ | $2.1115_{\pm0.097}$ | $1.2854_{\pm0.032}$ | $0.8773_{\pm0.056}$ | $1.3799_{\pm0.041}$ |
| | CIGIN | $0.7662_{\pm0.017}$ | $1.3649_{\pm0.021}$ | $0.5574_{\pm0.002}$ | $0.6383_{\pm0.005}$ | $0.7503_{\pm0.053}$ | $0.8642_{\pm0.012}$ |
| | CGIB | $0.8312_{\pm0.017}$ | $2.2118_{\pm0.024}$ | $0.3886_{\pm0.025}$ | $\underline{0.5476}_{\pm0.026}$ | $1.4525_{\pm0.013}$ | $0.7894_{\pm0.006}$ |
| | CMRL | $0.8063_{\pm0.012}$ | $2.1524_{\pm0.032}$ | $\underline{0.3777}_{\pm0.023}$ | $0.6672_{\pm0.013}$ | $1.4425_{\pm0.016}$ | $0.7894_{\pm0.002}$ |
| | RILOOD | $\mathbf{0.6784}_{\pm0.007}$ | $\mathbf{1.0780}_{\pm0.013}$ | $\mathbf{0.3660}_{\pm0.022}$ | $\mathbf{0.5209}_{\pm0.014}$ | $\mathbf{0.7001}_{\pm0.001}$ | $\mathbf{0.6991}_{\pm0.003}$ |

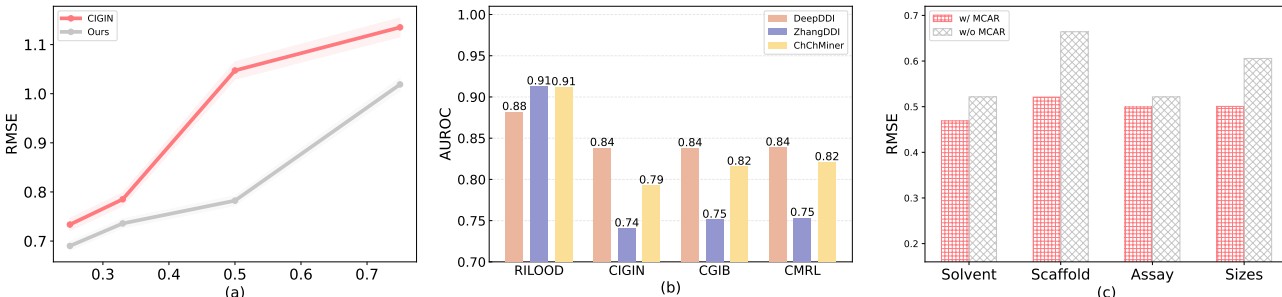

*Figure 3.* (a) Performance under different spurious correlation levels, where the strength of spurious correlation is defined as $d = \frac{\text{Number of samples with spurious features}}{\text{Total number of samples}}$. A higher $d$ in the training set indicates stronger spurious correlations with the underlying environments. (b) Results on three $DDI$ datasets with domain shifts, comparing our approach against three SOTA methods. (c) The impact of different interaction modes on the $CompSolv$ dataset, where w/ MCAR denotes a multi-granularity interaction mode and w/o MCAR indicates a node-level interaction mode.

As illustrated in Fig. 3(b), RILOOD consistently outperforms existing approaches under OOD conditions. This performance gain can be attributed to RILOOD's enhanced generalization capability, which facilitates effective knowledge transfer from known molecular interactions to structurally similar compounds and previously unseen scaffolds. Such transferability improves the model's robustness to distributional shifts, thereby ensuring adaptability across diverse molecular structures.

### 5.3. In-depth Analysis (RQ2)

To assess the contribution of each module, we conduct an ablation study by removing specific components: Multi-granularity Context-Aware Refinement (MCAR) trained on the downstream task (M); mutual information loss $\mathcal{L}_{MI}$ (Mi); conditional distribution modeling loss $\mathcal{L}_{MCVAE}$ (MC); and MCAR removal but all loss is used (w/o MCAR). The complete model is jointly trained using Eq. 13 (Ours). The results are summarized in Tab. 2. From Tab. 2, we observe the following: (1) Incorporating MCAR improves baseline performance, highlighting the importance of context-aware interactions in enhancing model robustness. (2) While conditional modeling significantly affects performance, the individual contributions of $\mathcal{L}_{MCVAE}$ and $\mathcal{L}_{MI}$ are smaller compared to joint training. (3) Removing MCAR leads to a performance drop; however, the model still surpasses the baseline due to the co-optimization of all losses.

**Feature Visualization.** To evaluate the effectiveness of MCAR, we used t-SNE to visualize molecular interactions in the best performing model and compare them to baselines. As shown in Fig. 4, (1) The solute in the test set originated from different distributions than the training set, demonstrating the distribution shift; (2) MCAR enhances feature diversity, improving molecule interaction modeling; and (3) MCAR captures domain-invariant features, boosting generalization to unseen domains. These results confirm our

*Table 2.* Ablation study on CompSolv-∗ and MNSolv-∗ by RMSE. We show the results of our method that performs best among baselines on all CompSolv-∗ and MNSolv-∗ datasets, for comparison.

| Method | CompSolv↓ | | MNSolv↓ | |
|---|---|---|---|---|
| | Solvent | Scaffold | Solvent | Scaffold |
| Baseline [B] | $0.5215_{\pm 0.007}$ | $0.6383_{\pm 0.011}$ | $0.7662_{\pm 0.016}$ | $1.2648_{\pm 0.018}$ |
| B + ERM loss [E] | $0.4864_{\pm 0.023}$ | $0.5919_{\pm 0.012}$ | $0.7263_{\pm 0.026}$ | $1.3478_{\pm 0.011}$ |
| B + MCAR [M] | $0.4914_{\pm 0.004}$ | $0.5842_{\pm 0.003}$ | $0.7115_{\pm 0.003}$ | $1.2191_{\pm 0.012}$ |
| M + $\mathcal{L}_{MI}$ [Mi] | $0.5196_{\pm 0.003}$ | $0.5444_{\pm 0.022}$ | $0.7279_{\pm 0.002}$ | $1.2005_{\pm 0.003}$ |
| Mi + $\mathcal{L}_{MCVAE}$ [MC] | $0.4753_{\pm 0.023}$ | $0.5351_{\pm 0.027}$ | $0.7026_{\pm 0.013}$ | $1.1329_{\pm 0.011}$ |
| w/o MCAR | $0.4834_{\pm 0.001}$ | $0.6641_{\pm 0.013}$ | $0.7285_{\pm 0.003}$ | $1.3929_{\pm 0.002}$ |
| Ours | $\mathbf{0.4689}_{\pm 0.006}$ | $\mathbf{0.5209}_{\pm 0.014}$ | $\mathbf{0.6784}_{\pm 0.007}$ | $\mathbf{1.0780}_{\pm 0.013}$ |

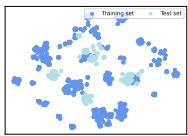 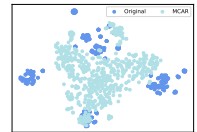 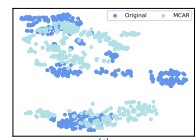

*Figure 4.* Visualization of the extracted features on training and validation set when the model achieves the best performance on the validation set. (a) The feature distribution of the training set and the test set; (b) Effect of MCAR on solute feature distribution; (c) Effect of MCAR on global feature (solute + solvent) distribution.

method's robustness against distribution shifts.

## 6. Conclusion

In this work, we propose a Relational Invariant Learning framework to solve out-of-distribution in solvation free energy prediction. Three tailored modules are jointly optimized to train the model and learn the representation of invariant molecules in diverse environments. Mixup enhanced molecular representations are used for variational modeling of diverse environments, further capturing invariant interaction patterns through multi-granularity context-aware refinement strategy. Extensive experiments and theoretical analysis prove the superiority of our method.

## Impact Statement

This paper presents work whose goal is to advance the field of AI for Science. Our method may have some potential societal consequences, but we don't believe any need specific highlighting here.

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

# A. Proofs

In this section, we provides detailed proofs in Section 4.

## A.1. Proof for Bound of Mixup Representations

Let $H_1 \in \mathbb{R}^d$ be the representation of a solute graph $\mathcal{G}_1$, and $H_2 \in \mathbb{R}^d$ be the representation of a solvent graph $\mathcal{G}_2$ under environment $e$, both obtained via a GNN. We define the **mixup representation** as:

$$\tilde{H} = \lambda H_1 + (1 - \lambda)H_2, \quad \lambda \in [0, 1] \tag{14}$$

We assume the following: The predictive model $f : \mathbb{R}^d \to \mathcal{Y}$ is Lipschitz continuous with constant $L_f > 0$, i.e., for any $H, H' \in \mathbb{R}^d$,

$$\|f(H) - f(H')\|_{\mathcal{Y}} \leq L_f \cdot \|H - H'\|_2 \tag{15}$$

The loss function $\ell : \mathcal{Y} \times \mathcal{Y}_{\text{target}} \to \mathbb{R} \geq 0$ is upper-bounded by constant $B$:

$$0 \leq \ell(\mathcal{Y}_{\text{pred}}, \mathcal{Y}_{\text{true}}) \leq B, \quad \forall \mathcal{Y}_{\text{pred}}, \mathcal{Y}_{\text{true}} \tag{16}$$

Let $\mathcal{D}$ be the joint distribution over $(H_1, H_2, e_1, e_2)$, where $e_1, e_2 \in \mathcal{Y}_{\text{target}}$. Define the expected prediction error under mixup as:

$$\mathcal{E}_{\text{mixup}}(\lambda) := \mathbb{E}_{(H_1, H_2, e_1, e_2) \sim \mathcal{D}} \left[ \ell(f(\tilde{H}), e) \right], \tag{17}$$

where $e = \lambda e_1 + (1 - \lambda)e_2$ denotes environment, i.e., the soft target label. According to the information-theoretic generalization bounds based on Fano's inequality, we obtain:

$$\mathcal{E}_{\text{mixup}}(\lambda) \geq c_1 \cdot \left( H(\mathcal{Y}_{\text{target}}) - I(\tilde{H}; \mathcal{Y}_{\text{target}}) \right) \tag{18}$$

where $H(\mathcal{Y}_{\text{target}})$ is the entropy of the target $\mathcal{Y}_{\text{target}}$, and $I(\tilde{H}; \mathcal{Y}_{\text{target}})$ is the mutual information between the representation $\tilde{H}$ and the soft label. Constant $c_1 > 0$ depends on the task structure and loss smoothness.

**Generalization via Nuisance Information.** From a representation learning perspective, generalization error can be bounded by:

$$\mathcal{E}_{\text{mixup}}(\lambda) \leq \mathcal{E}^* + c_2 \cdot \sqrt{I(\tilde{H}; C_{\text{nuisance}})}, \tag{19}$$

where $C_{\text{nuisance}}$ denotes spurious nuisance factors, $c_2 > 0$ depends on $L_f$ and $B$, and $\mathcal{E}^*$ denotes the optimal lower bound of the achievable generalization error.

Therefore, we obtain:

$$c_1 \cdot \left( H(Y) - I(\tilde{H}; Y) \right) \leq \mathcal{E}_{\text{mixup}}(\lambda) \leq \mathcal{E}^* + c_2 \cdot \sqrt{I(\tilde{H}; C_{\text{nuisance}})} \tag{20}$$

These derivations indicate a fundamental trade-off in mixup-based CVAE learning: while mixup reduces nuisance information through distributional smoothing, it must preserve sufficient task-relevant information to avoid performance degradation. This analysis provides a theoretical justification for using mixup representations in conditional generative models under structured environments.

## A.2. Proof for Equation 6

For a given mixed representation $\tilde{H}_1$ and environment $e$, the loss function is defined as:

$$L(\theta, \phi; \tilde{H}_1, e) = \mathbb{E}_{q_\phi(z|\tilde{H}_1, e)}[\log p_\theta(\tilde{H}_1 \mid z, e)] - D_{KL}(q_\phi(z \mid \tilde{H}_1, e) \| p_\theta(z \mid \tilde{H}_1)) \tag{21}$$

Here, the learning objective is to find the optimal parameter set $\theta$ that maximizes the log-likelihood $\log p_\theta(\tilde{H}_1 \mid e)$. Since directly computing the true posterior $p_\theta(z \mid \tilde{H}_1, e)$ is intractable, we approximate it using an auxiliary variational distribution

$q_\phi(z \mid \tilde{H}_1, e)$. By minimizing the KL divergence between $q_\phi(z \mid \tilde{H}_1, e)$ and $p_\theta(z \mid e)$, we derive a tractable Evidence Lower Bound (ELBO) formulation:

$$\max_{\theta,\phi} \mathbb{E}_{\mathcal{G}_1 \sim D} \left[ \mathbb{E}_{q_\phi(z \mid \tilde{H}_1, e)} \left[ \log p_\theta(\tilde{H}_1 \mid z, e) \right] \right] \quad \text{s.t.} \quad D_{KL}\left( q_\phi(z \mid \tilde{H}_1, e) \| p_\theta(z \mid \tilde{H}_1) \right) < \delta \tag{22}$$

where $\delta$ is a threshold that ensures the learned latent representation $z$ remains close to the true underlying data distribution. We define $q_\phi(z \mid \tilde{H}_1, e)$ as the recognition model and $p_\theta(\tilde{H}_1 \mid e, z)$ as the generative model. By leveraging approximate posterior inference and the reparameterization trick, the prior is able to effectively capture environmental information from the posterior, thereby improving posterior alignment.

Using the ELBO decomposition, we express the log-likelihood as:

$$\log p_\theta(\tilde{H}_1 \mid e) = -D_{KL}(q_\phi(z \mid \tilde{H}_1, e) \| p_\theta(z \mid \tilde{H}_1)) + \mathbb{E}_{q_\phi(z \mid \tilde{H}_1, e)} \left[ \log p_\theta(\tilde{H}_1 \mid z, e) \right] \tag{23}$$

where $D_{KL}(\cdot \| \cdot)$ denotes the Kullback-Leibler divergence between two distributions.

For regression-based solvation property prediction, we introduce an uncertainty constraint on RMSE, modifying the reconstruction term as follows:

$$\begin{aligned}
\mathcal{L}_{\text{MCVAE}}(\theta, \phi; \tilde{H}_1, e) = &-D_{KL}\left( q_\phi(z \mid \tilde{H}_1, e) \| p_\theta(z \mid \tilde{H}_1) \right) \\
&+ \frac{1}{N} \sum_{i=1}^{N} \left[ \frac{1}{\sigma(\tilde{H}_1)^2} \| z - \tilde{H}_1 \|^2 + \log \sigma(\tilde{H}_1)^2 \right]
\end{aligned} \tag{24}$$

This term ensures that the variance of the latent variable $\sigma(\tilde{H}_1)^2$ is explicitly accounted for, promoting robust representation learning and reducing overfitting to noise in the data.

### A.3. Details of Assumption 4.1

We define two key properties, *invariance* and *sufficiency*, which ensure the generalizability of molecular relational learning across different environments. For given OOD environment $e'$, we define invariance and sufficiency as:

(1) *Invariance Property* : $\forall e, e' \in supp(\varepsilon)$, $p(\mathcal{Y}^e \mid H^e, e) = p(\mathcal{Y}^{e'} \mid H^{e'}, e')$, where $H^i = \Phi(\mathcal{G}_1^i, \mathcal{G}_2^i)$ denotes molecular pairs representations, $H^e = \Phi(\mathcal{G}_1^e, \mathcal{G}_2^e)$, $H^{e'} = \Phi(\mathcal{G}_1^{e'}, \mathcal{G}_2^{e'})$;

(2) *Sufficiency Property* : $\mathcal{Y} = \Phi(\mathcal{G}_1^e, \mathcal{G}_2^e) + \epsilon$, where $\Phi$ is a predictor, $\epsilon$ is a random noise.

If the following conditions hold:

- Conditional Independence:
$$\Phi(\mathcal{G}_1, \mathcal{G}_2) \perp (\mathcal{G}_1, \mathcal{G}_2) \setminus \Phi(\mathcal{G}_1, \mathcal{G}_2), \tag{25}$$
  which ensures that the learned representation is conditionally independent of the remaining molecular graph information.

- For all environments $e \in \text{supp}(\mathcal{E})$, there exists OOD environment $e' \in \text{supp}(\mathcal{E})$ satisfy:
$$p_{e'}((\mathcal{G}_1, \mathcal{G}_2), \mathcal{Y}) = p_{e'}(\Phi(\mathcal{G}_1, \mathcal{G}_2), \mathcal{Y}) p_{e'}((\mathcal{G}_1, \mathcal{G}_2) \setminus \Phi(\mathcal{G}_1, \mathcal{G}_2)), \tag{26}$$
  which implies that the joint distribution factorizes into invariant representations and residual molecular information.

- Invariance of representation across environments:
$$p_{e'}(\Phi(\mathcal{G}_1, \mathcal{G}_2)) = p_e(\Phi(\mathcal{G}_1, \mathcal{G}_2)). \tag{27}$$

This guarantees that the representation distribution remains stable across environments $e$ and $e'$, reinforcing the assumption of invariant learning.

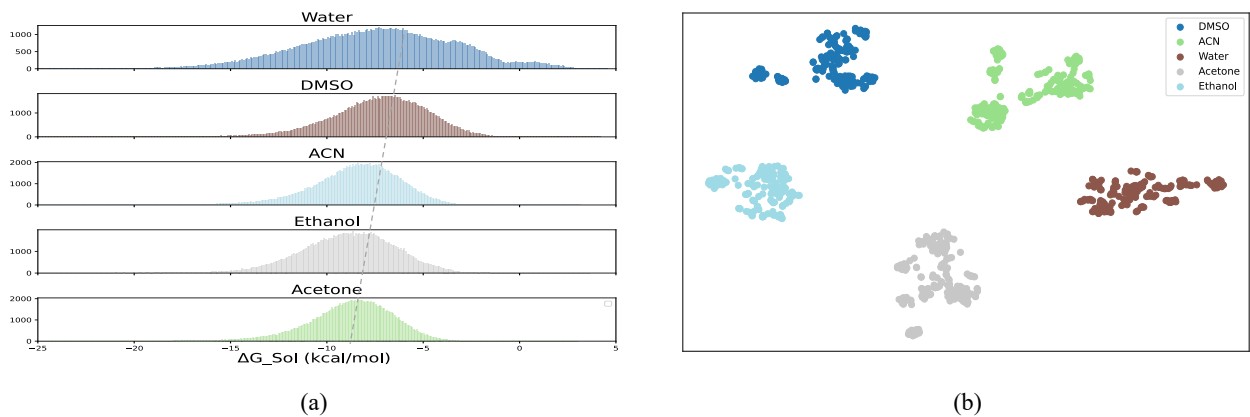

*Figure 5.* (a) The distribution of property labels of solute molecules in five solvents (water, acetone, ethanol, ACN, and DMSO) in the QM9Solv dataset. (b) t-SNE visualization of the representation distributions of solute molecules in the QM9Solv dataset across the same five solvents.

## B. Details on Datasets

We provide more details about the benchmark datasets that we use in our experiments. Statistics of the datasets are provided in Table 3.

- **MNSolv** dataset includes 3,037 experimentally measured free energies of solvation or transfer for 790 distinct solutes and 92 different solvents. In this work, we remove the charged ions, and we consider 2,577 combinations of 528 unique solutes and 104 solvents, including mixed solvents.

- **QM9Solv** dataset includes solvation energies for 130,258 molecules from the QM9 dataset, calculated in 5 solvents (acetone, ethanol, acetonitrile, dimethyl sulfoxide, and water) using an implicit solvent model.

- **CompSolv** dataset is designed to illustrate the impact of hydrogen-bonding association effects on solvation energies. It includes 3,548 pairings of 442 distinct solutes and 259 solvents, based on prior studies. The dataset also contains the original assay attribution of the data, which we have divided it into OOD settings.

- **CombiSolv** integrates all data from MNSol, FreeSolv, CompSolv, and Abraham. After we removed the outliers, there were 1,000,000 entries, including 11,029 solutes and 284 solvents.

- **MolMerger** compiled a dataset of 6,975 entries from three sources: BigSolDB (Krasnov et al., 2023), BNNLabs Solubility (Boobier et al., 2020), and ESOL (Delaney, 2004). The data obtained from BigSolDB required cleaning to remove erroneous or unreliable entries.

- **Abraham** dataset, compiled by the Abraham research group at University College London, comprises 6,091 pairings of 1,038 unique solutes and 122 solvents, building on previous studies.

- **ZhangDDI** (Zhang et al., 2017b), **ChChMiner** (Marinka Zitnik et al., 2018) and **DeepDDI** (Ryu et al., 2018) are datasets focused on drug–drug interaction prediction, widely used for evaluating molecular relation learning models.

### B.1. Data splitting.

To evaluate the OOD generalization performance of molecule relational learning models, we employed both random scaffold splitting and random solvent splitting strategies.

**Random Scaffold Splitting** (Bemis & Murcko, 1996) is a method where the dataset is grouped and randomly split based on molecular scaffolds. Specifically, the Bemis-Murcko was used to extract the scaffold structure of each molecule, and molecules with the same scaffold were grouped together. These scaffold groups were then randomly shuffled and split into training, validation, and test sets according to a predefined ratio (e.g., 8:1:1). This approach ensures that molecules with the same scaffold appear exclusively in one set, allowing the evaluation of the model's generalization ability to unseen scaffolds.

*Table 3.* Details about the datasets used in OOD experiments.

| Dataset | G1 | G2 | Target | #G1 | #G2 |
|---------|------|--------|------------------|--------|------|
| MNSolv | Solute | Solvent | Solvation Energy | 528 | 104 |
| QM9Solv | Solute | Solvent | Solvation Energy | 130625 | 5 |
| CompSolv | Solute | Solvent | Solvation Energy | 442 | 259 |
| CombiSolv | Solute | Solvent | Solvation Energy | 11029 | 284 |
| MolMerger | Solute | Solvent | LogS | 2699 | 13 |
| Abraham | Solute | Solvent | Solvation Energy | 1038 | 122 |

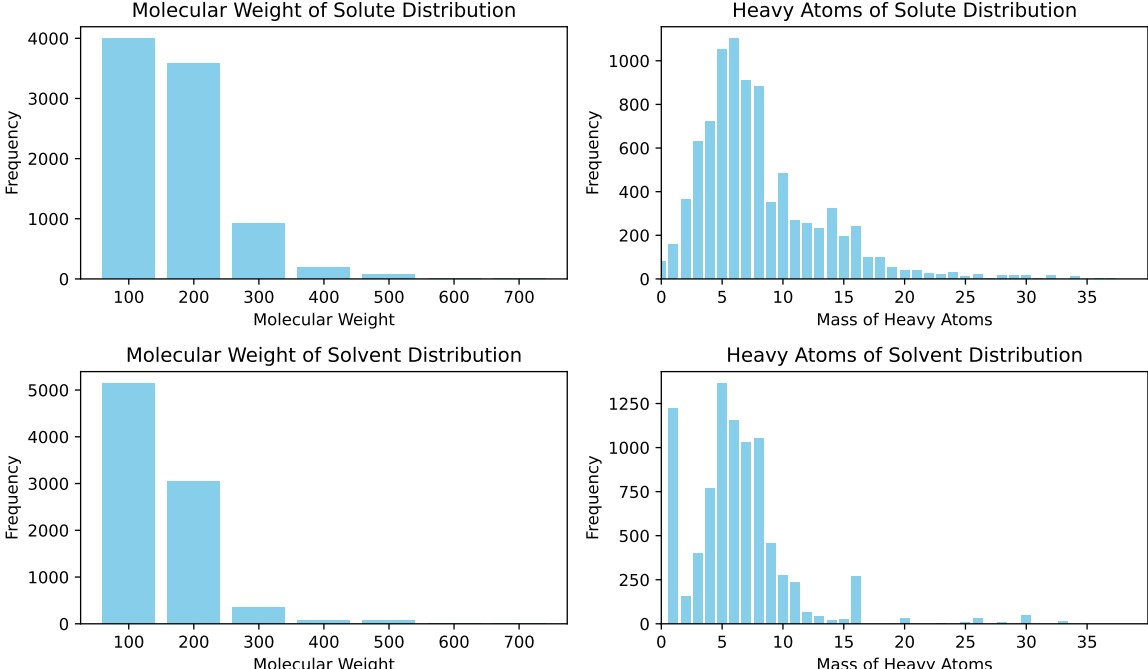

*Figure 6.* Histograms of the molecular weight and heavy atomic distribution of solutes and solvents in the $\mathrm{CombiSolv}$ dataset. Based on the solvent solute ratios in Table 5, we observe that the solutes and solvents in this dataset are mainly composed of small molecules and exhibit a long-tail distribution in molecular weight and heavy atom number. The average molecular weight of the solvent is lower than that of the solute and may be mainly common small molecule solvents. This distribution can pose a challenge to the generalization ability of the model, especially when working with large molecules or complex solvents.

**Random Solvent Splitting** is a dataset partitioning strategy based on solvents, similar to random scaffold splitting. Solvent-domain shift considers the scenario in which the solute distribution $p(\mathcal{G}_1|\mathcal{G}_2)$ is shifted across solvent splits. Here, solute molecules are grouped according to their corresponding solvents, and molecules with the same solvent are assigned to the same group. These solvent groups are then randomly shuffled and divided into training, validation, and test sets in a fixed ratio. Each solvent is assigned to only one set, ensuring that the solvents in the test set are entirely unseen during training. This strategy aims to evaluate the model's performance in predicting molecular properties in new solvent environments and its adaptability to out-of-distribution solvents.

## C. More Experiments and details

### C.1. Implementation and Optimization Details.

The proposed method is implemented on a single NVIDA 3090 GPU with PyTorch. Following the CIGIN(Pathak et al., 2020), we use the same 3-layer GCN and MPNN as feature extractor for solute molecule and solvent molecule, respectively. More details about backbone can be found in Sec.3.1. During the training, the solute features were incorporated with node interaction features, which is the dot-production similarity between solute node features and solvent node features. Here, we

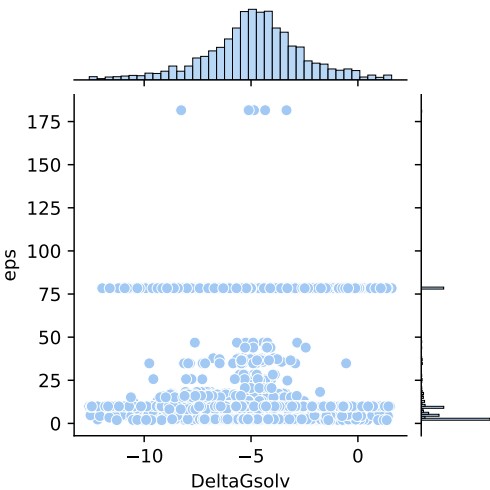

*Figure 7.* Spurious correlation from dielectric constant $eps$.

*Table 4.* Performance on molecular interaction prediction task (regression) in terms of RMSE.

| Model | Chromophore | | | MNSolv | QM9Solv | FreeSolv | CompSolv | Abraham | CombiSolv |
|---|---|---|---|---|---|---|---|---|---|
| | **Absorption** | **Emission** | **Lifetime** | | | | | | |
| GCN | $25.75_{\pm1.48}$ | $31.87_{\pm1.70}$ | $0.866_{\pm0.015}$ | $0.675_{\pm0.021}$ | $1.317_{\pm0.011}$ | $1.192_{\pm0.042}$ | $0.389_{\pm0.009}$ | $0.738_{\pm0.041}$ | $0.672_{\pm0.022}$ |
| GIN | $24.92_{\pm1.67}$ | $32.31_{\pm0.26}$ | $0.829_{\pm0.027}$ | $0.669_{\pm0.013}$ | $1.274_{\pm0.017}$ | $1.015_{\pm0.041}$ | $0.331_{\pm0.016}$ | $0.648_{\pm0.024}$ | $0.595_{\pm0.014}$ |
| GAT | $26.19_{\pm1.44}$ | $30.90_{\pm1.01}$ | $0.859_{\pm0.016}$ | $0.731_{\pm0.007}$ | $1.305_{\pm0.021}$ | $1.280_{\pm0.049}$ | $0.387_{\pm0.010}$ | $0.798_{\pm0.038}$ | $0.662_{\pm0.021}$ |
| CIGIN | $19.32_{\pm0.35}$ | $25.09_{\pm0.32}$ | $0.804_{\pm0.010}$ | $0.607_{\pm0.024}$ | $0.592_{\pm0.023}$ | $0.905_{\pm0.014}$ | $0.308_{\pm0.018}$ | $0.411_{\pm0.008}$ | $0.451_{\pm0.009}$ |
| CGIB | $18.11_{\pm0.21}$ | $23.90_{\pm0.12}$ | $0.771_{\pm0.020}$ | $0.538_{\pm0.015}$ | $0.549_{\pm0.026}$ | $0.852_{\pm0.032}$ | $0.276_{\pm0.013}$ | $0.390_{\pm0.016}$ | $0.422_{\pm0.005}$ |
| CMRL | $17.93_{\pm0.31}$ | $24.30_{\pm0.22}$ | $0.776_{\pm0.007}$ | $0.551_{\pm0.017}$ | $0.288_{\pm0.013}$ | $\mathbf{0.815}_{\pm0.046}$ | $0.255_{\pm0.011}$ | $0.374_{\pm0.011}$ | $0.421_{\pm0.008}$ |
| RILOOD | $\mathbf{17.20}_{\pm0.12}$ | $\mathbf{23.61}_{\pm0.21}$ | $\mathbf{0.706}_{\pm0.015}$ | $\mathbf{0.489}_{\pm0.009}$ | $\mathbf{0.246}_{\pm0.016}$ | $0.823_{\pm0.017}$ | $\mathbf{0.242}_{\pm0.018}$ | $\mathbf{0.309}_{\pm0.013}$ | $\mathbf{0.292}_{\pm0.009}$ |

using graph-level solute features and solvent features as input in our method. We select 168 for the dimension ($d_z$) of latent variables. The learning rate was decreased on plateau by a factor of $10^{-3}$ from $10^{-3}$ to $10^{-5}$.

### C.2. Synthetic data experiments.

We first consider the distribution shift caused by polarity bias w.r.t. eps. The invariant feature is $\widehat{H}_{inv} \in \mathbb{R}$, where $p(\mathcal{Y}|\widehat{H}_{inv})$ is a constant, indicating a stable correlation between $\mathcal{Y}$ and $\widehat{H}_{inv}$. Our goal is to learn a model that relies solely on $\widehat{H}_{inv}$. We use $eps$ to control the degree of spurious correlation. Fig. 7 illustrates a spurious correlation arising from the dielectric constant, which plays a crucial role in determining the polarity of the solvent. As shown in Fig. 7, the correlation of molecular pairs and label $\mathcal{Y}$ with $eps$=78 is unstable, i.e., $p(\mathcal{Y}|H_1)$ is unstable, $p(\mathcal{Y}|\widehat{H}_{inv})$ is stable.

### C.3. Generalization Analysis.

Table 4 presents the RMSE performance of various models on molecular interaction prediction tasks under IID settings. RILOOD consistently achieves the lowest RMSE across most datasets, demonstrating superior predictive capability. Traditional graph-based models (GCN, GIN, GAT) perform worse, particularly in QM9Solv, FreeSolv, and CombiSolv, indicating limitations in capturing complex molecular interactions. More advanced models show notable improvements, yet RILOOD surpasses them in nearly all cases, with significant gains in QM9Solv, CompSolv, and CombiSolv. In Chromophore datasets, RILOOD also achieves the best results, particularly in Absorption and Emission, highlighting its strength in modeling spectral properties. The Multi-granularity Context-Aware Refinement (MCAR) and Mixup-based CVAE (MCVAE) mechanisms likely enhance feature extraction and generalization. Although RILOOD slightly underperforms CGIB in FreeSolv, its overall performance validates its effectiveness in molecular representation learning, even under IID conditions.

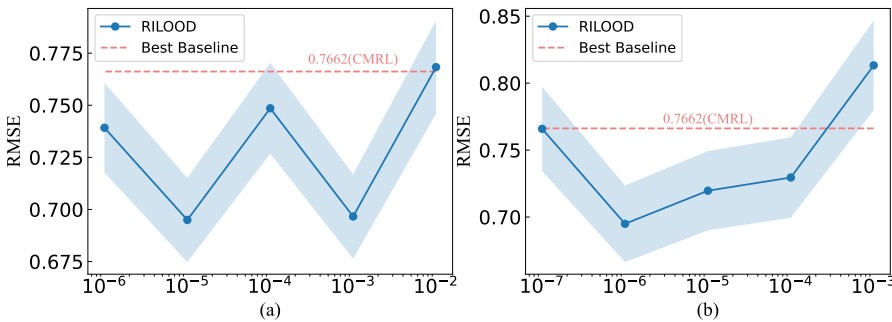

*Figure 8.* Sensitivity analysis of the hyperparameter (a) $\alpha$ and (b) $\beta$ on $\mathrm{CompSolv}$ datasets. The solid line shows the average RMSE in the testing stage and the light blue area represents standard deviations. The dashed line represents the average RMSE of the best-performed baseline.

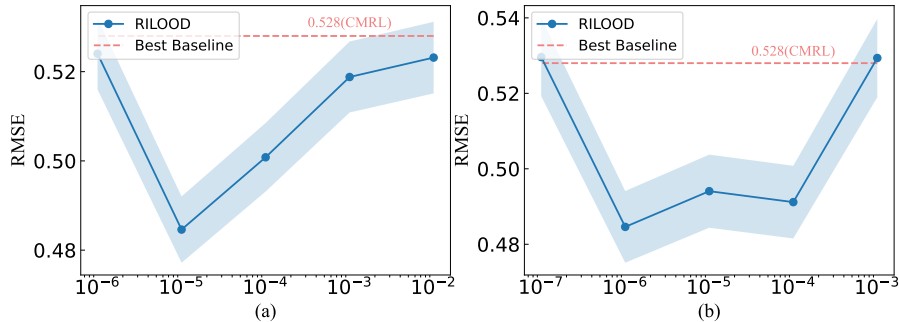

*Figure 9.* Sensitivity analysis of the hyperparameter (a) $\alpha$ and (b) $\beta$ on $\mathrm{MNSolv}$ datasets. The solid line shows the average RMSE in the testing stage and the light blue area represents standard deviations. The dashed line represents the average RMSE of the best-performed baseline.

## C.4. Hyperparameter Sensitivity Analysis

We perform a sensitivity analysis on the hyperparameters $\alpha$ and $\beta$, which govern the trade-off among the loss components in Eq. 13. In practical settings, accurately approximating the true posterior is non-trivial, often resulting in a reconstruction loss that dominates the supervised loss by several orders of magnitude. To mitigate this imbalance and stabilize optimization, we systematically vary $\alpha$ within $\{10^{-7}, 10^{-6}, 10^{-5}, 10^{-4}, 10^{-3}\}$ and $\beta$ within $\{10^{-8}, 10^{-7}, 10^{-6}, 10^{-5}, 10^{-4}\}$. Experiments are conducted on the $\mathrm{MNSolv}$ and $\mathrm{CompSolv}$ datasets, chosen for their diversity and representativeness.

The results, shown in Fig.8 and Fig.9, demonstrate that the choice of $\alpha$ and $\beta$ critically affects the balance between environmental modeling and invariant representation learning. Notably, higher values of $\alpha$ tend to yield significant performance improvements. These findings underscore the importance of tuning these hyperparameters for optimal performance. Following established practice, we report the best-performing configuration along with its standard deviation.

## D. Limitations and Future Directions

While we introduce a molecular pair invariant graph learning approach based on auxiliary information for environment partitioning, achieving strong OOD generalization, certain limitations remain. Since solvent polarity labels are difficult to obtain, auxiliary information can serve as a proxy label for environment partitioning.

This study examines spurious correlations by considering a single spurious factor; however, in reality, multiple spurious associations may exist, complicating the distinction between false and invariant patterns. The influence of different spurious factors varies, impacting model performance in distinct ways.

Additionally, in the era of large-scale models, increasing data volume improves OOD generalization. However, in domain-specific applications, prioritizing broad generalization may reduce accuracy in specialized fields. This trade-off arises

because narrow distributions focus on high-probability regions, often overlooking tail samples, leading to poor generalization in rare cases. Bridging this gap between broad generalization and domain-specific precision presents an important research challenge.

