# OpenReview forum: "Relational Invariant Learning for Robust Solvation Free Energy Prediction"
_ICML.cc/2025/Conference — ICML 2025 spotlightposter_

### Official Review · Reviewer_b6ge · 2025-03-10

**Overall Recommendation:** 3

**Summary:**

The paper proposes Relational Invariant Learning framework for solvation free energy prediction. RILOOD consists of three key components: a mixed conditional modeling module to integrate data from different environments, a multi-granularity refinement strategy for context-aware representation learning, and an invariant learning module to identify robust patterns that generalize to unseen environments. Experimental results demonstrate that RILOOD significantly outperforms existing SOTA methods across various datasets, showcasing its effectiveness in capturing complex molecular interactions. The paper highlights the challenges of OOD scenarios and suggests future research directions to balance broad generalization with domain-specific precision.

**Claims And Evidence:**

Yes.

**Essential References Not Discussed:**

No

**Experimental Designs Or Analyses:**

This paper conducted a variety of experiments to illustrate the effectiveness of the method, including comparative experiments, ablation experiments, and super-parameter experiments, and gave detailed analyses.

**Methods And Evaluation Criteria:**

I don’t think the baseline comparison is strong enough due to the insufficiency of OOD datasets. Compared to Table 4, which includes a broader range of datasets, Table 1 only compares three datasets. The authors should consider including more datasets to better illustrate the effectiveness of the method.

**Other Comments Or Suggestions:**

- The appendix in the capital of Table 1 is incorrect.
- Fig.1(b), the abscissa SMILES should be the corresponding name for ease of understanding.
- There is an overlap in the data on the Fig.3 (b).

**Other Strengths And Weaknesses:**

**Strengths**
- The motivation of the paper is clearly defined.
- This paper is well-written and effectively organized.
- The topics discussed are very important in practical application.
- The experiments show that RILOOD achieves robust solvation free energy prediction and outperforms baseline models under distribution shifts.

**Weaknesses**
- Personally, I do not find CVAE to be particularly innovative, as it is often used in molecular generation[1].
- This paper is only focused on the task of solvation free energy prediction, and the method is difficult to extend to general molecular property prediction or DDI tasks.
- Authors should add case studies illustrating generalization performance in practical applications.

[1] Lim, Jaechang, et al. "Molecular generative model based on conditional variational autoencoder for de novo molecular design." Journal of cheminformatics 10 (2018): 1-9.

**Questions For Authors:**

- What does the molecular interaction in Fig.1(a) mean by arrows of the same color? How to account for interaction invariance?
- Why is w/o *MCAR* better than *B+MCAR[M]* in Table 2?
- What is the difference between c_r, C, c_1, c_2, C?

**Relation To Broader Scientific Literature:**

In this paper, the authors extend the traditional molecular property prediction to solvent environment, and consider the role of solvents, which is helpful to further expand the practical application of existing AI for science.

**Theoretical Claims:**

I carefully reviewed the proof in the appendix, and the proof of theoretical claims is generally well structured, but there are some points that require further clarification and rigor, and addressing these issues will increase the credibility of the theoretical framework.

- The theoretical rationale for the uncertainty constraint of RMSE in A.1 is interesting, but the proof provided lacks sufficient detail. It would be helpful to have a more comprehensive analysis of how this constraint affects the representation of learning, and to provide formal evidence of its effectiveness in improving the robustness of the model.

- For proof A.2, acknowledging any potential limitations or assumptions that may not hold in certain scenarios would provide a more balanced view.

---

> ### Author Rebuttal · Authors · 2025-04-01
>
> Dear reviewer b6ge:
>
> Thank you for your thoughtful suggestions and questions. We have provided point-by-point answers to each weakness and question.
>
> ## Weakness
> **W1. Personally, I do not find CVAE to be particularly innovative.**
>
> While we understand your perspective on the novelty of the CVAE framework, our focus in this paper is not solely on theoretical innovation or technological advancement. Instead, we aim to address a valuable domain-specific problem: solvent generalization in practical applications. We believe that our work is pioneering in its application of CVAE to the context of solvent generalization, which has significant implications for real-world scenarios. By concentrating on this specific challenge, we provide insights and solutions that can enhance the predictive capabilities of models in the field of solvation properties.
>
> **W2. This paper is only focused on the task of solvation free energy prediction, and the method is difficult to extend to general molecular property prediction or DDI tasks.**
>
> As we mentioned in the limitation, the types of environments don't increase indefinitely. Considering the scalability and computational cost of the model, when there are too many types of solvents, clustering can be performed first, and then the pseudo-labels of the clustering can be conditionally modeled, so we think that our approach can be extended to more task types in the future.
>
> **W3. More case study:**
> We've added some solvent-holdout test cases to verify real-world cases. Due to time constraints, we only selected a few cases. As can be seen from the case studies, the prediction error is still low for different types of solvents, including alkanes, alcohols, and olefins.
>
> Table. The results of RILOOD on solvents holdout test.
> |  Solvent  | RMSE |
> |-------|-------|
> | Tetraglyme  |  0.1004  |
> | Pentadecane |  0.1100  |
> | Tetradecane |  0.2462  |
> | 1-Dodecanol |  0.2315  |
> | 1-bromonaphthalene | 0.7796 |
> | Tetraglycol |  0.2015  |
>
> **Minor issue.** We’ve corrected errors and updated the manuscript.
>
> ## Question:
>
> **Q2. Why is w/o MCAR better than B+MCAR[M] in Table 2?**
>
> w/o MCAR is better than B+MCAR[M] and only performs on one dataset, which may be due to the fact that w/o MCAR only removes the interactive module and the objective function does not change, which leads to the model being reverse-optimized.
>
> **Q3. What is the difference between $c^r$, $C$, $c_1$, $c_2$?**
>
> $C$ is the collection of all solvents, $c^r$ refers to a specific class of solvents, $c_1$ refers to the type of solute, and $c_2$ refers to the type of solvent
>
> We hope that our response will address your concerns and better clarify the contribution and value of our work.

---

> > ### Comment · Reviewer_b6ge · 2025-04-07
> >
> > Thanks so much for your thorough rebuttal. It addressed the majority of my concerns, and I'm willing to raise my score.

---

> > > ### Author Response · Authors · 2025-04-08
> > >
> > > Thank you for your insightful review. We are very grateful for your meticulous review and constructive criticism of our work.

---

### Official Review · Reviewer_9m7P · 2025-03-10

**Overall Recommendation:** 4

**Summary:**

This paper investigates the challenge of out-of-distribution generalization across different environments in molecular solvation free energy prediction and introduces the RILOOD framework. RILOOD integrates mixup-based conditional modeling, a multi-granularity refinement strategy, and an invariant relational learning module to alleviate the limitations of traditional methods that overly rely on core substructures. Experimental results demonstrate that RILOOD significantly outperforms existing methods on multiple benchmark datasets.

**Claims And Evidence:**

Some of the paper's claims have minor issues. The difference among environments, environmental conditions, and solvent categories is not clearly expressed.

**Essential References Not Discussed:**

I believe that essential references have been sufficiently discussed.

**Experimental Designs Or Analyses:**

1. The proposed method is evaluated using RMSE, MAE, AUROC, and Accuracy, with experiments conducted on both real-world and synthetic datasets.

2. t-SNE is used to visualize molecular interactions in the best performing model.

**Methods And Evaluation Criteria:**

The paper evaluates the effectiveness of the model and improving generalization in OOD scenarios. Extensive experiments and theoretical analysis prove the superiority of the method.

**Other Comments Or Suggestions:**

1. Lines 161-164 and 263-266 are blank.

2. There are too many variables, making the paper somewhat difficult to read.

3. In Eq. (4), $G$ should be $\mathcal{G}$.

4. There is a missing cite at line 332 ("Appendix ??").

**Other Strengths And Weaknesses:**

Strengths
1. Figure 1 clearly presents the motivation of the work.

2. The experiments include numerous visualizations that further validate the paper’s claims.

3. The performance improvements are significant.

Weaknesses
1. RILOOD integrates several modules to capture complex molecular interactions, but this results in a relatively complex model structure.

2. The distinction between "surrounding environments" and "support environments" is unclear.

3. Some variables are repeatedly defined (e.g., "K" is used for both surrounding environments and attention keys, "c" represents both coarse-grained conditions and environmental conditions); please check and reduce this confusion.

4. Both Sections 3.1 and 3.2 include a problem formulation, which is confusing.

**Questions For Authors:**

Please refer to the above comments.

**Relation To Broader Scientific Literature:**

This work is important. Predicting the solvation free energy of molecules in out-of-distribution (OOD) scenarios has greater practical value compared to identically distributed scenarios.

**Theoretical Claims:**

The paper appears to provide sufficient theoretical analysis.

---

> ### Author Rebuttal · Authors · 2025-04-01
>
> Dear reviewer 9m7P:
>
> Thank you for your thoughtful suggestions and questions.We have provided point-by-point answers to each weakness and question.
>
> **W1. RILOOD integrates several modules to capture complex molecular interactions, but this results in a relatively complex model structure:**
>
> We understand your concerns regarding the complexity of the model. We chose this complex model structure because it allows us to better capture the intricate interactions between molecules, which is crucial for improving the accuracy and reliability of our predictions.
>
> Although the model structure is relatively complex, we believe that this complexity enables the model to gain a more comprehensive understanding of molecular characteristics, thereby providing better performance in practical applications. Additionally, we have also considered interpretability in our model design and have made efforts to ensure that each module of the model has a clear function. We are also considering possible simplifications in future work to reduce complexity while maintaining model performance. We would greatly appreciate any further suggestions you may have regarding this.
>
> **W2. The distinction between "surrounding environments" and "support environments" is unclear:**
>
> "Surrounding environments" and "support environments" mean the same thing. We'll change it to a unified expression.
>
> **W3. Symbols are defined repeatedly:** Thank you for your corrections, and we will double-check for further proofreading.
>
> **W4. Both Sections 3.1 and 3.2 include a problem formulation, which is confusing:**
>
> The problem formulation in 3.1 is the molecular interaction prediction task in general, mainly under the IID setting; The problem formulation in 3.2, the purpose of OOD setting is to emphasize the difference between the distribution of the training set and the test set. We'll update the language to make it easier to understand.
>
> **Minor issues:** We’ve corrected errors and updated the description in the paper.
>
> Thank you for your valuable comments, and we hope that our response will address your concerns and better clarify the contribution and value of our work.

---

> > ### Comment · Reviewer_9m7P · 2025-04-08
> >
> > Thank you very much for your detailed and thoughtful rebuttal. It effectively addressed most of my concerns, and based on this, I will revise my rating upward.

---

### Official Review · Reviewer_zysq · 2025-03-11

**Overall Recommendation:** 4

**Summary:**

In this paper, the authors presents the Relational Invariant Learning framework (RILOOD) to improve OOD generalization in solvation free energy prediction. RILOOD learns invariant molecular representations in varied environments and applies mixed-enhanced molecular features for modeling environmental diversity. Extensive experiments demonstrate its superiority over existing methods under various distribution shifts.

**Claims And Evidence:**

Yes. Claims made in the submission supported by clear and convincing evidence.

**Essential References Not Discussed:**

No.

**Experimental Designs Or Analyses:**

The authors provide a large number of experiments to illustrate the advanced nature of the proposed method, and the experimental analysis is reasonable.

**Methods And Evaluation Criteria:**

The authors provide a wealth of benchmarks to prove the superiority of the proposed method, and the evaluation method is effective.

**Other Comments Or Suggestions:**

The writing is not rigorous, and it is repeated in many places。
- What does multiple environments in line 163 $\varepsilon$ mean? How it is and the 115 line of $\varepsilon$ is clearly not a representation.
- Is the $l$ in "drawing samples $z^{(l)} (l = 1, 2, ..., l)$" the same as the $l$ in "$h_{l+1} = PReLU(W_lh_l + b_l)$" in Equtation 10? $l$ is the same as $l$ in loss, it is recommended to change the symbol.

**Other Strengths And Weaknesses:**

**Strengths**
1. The author offers a new invariant learning method, which, as far as I know, is novel
2. Compared with the SOTA method, the experimental results show that the proposed method performs well.
3. The author provides a theoretical proof.

**Weaknesses**
1. Although theoretical support is provided, there is no innovation in theory
2. Figure 1 is intended to illustrate the invariance of interactions, but (a) the illustration is not easy to understand.
3. There is no clear definition of what invariant interaction patterns are.
4. Since obtaining explicit environment labels for solute-solvent pairs is often infeasible. So how is this method obtained?

**Questions For Authors:**

1. Could you provide real-world examples to further illustrate the OOD generalization of RILOOD?
2. About tsne, MCAR how to enhance feature diversity?

**Relation To Broader Scientific Literature:**

No.

**Theoretical Claims:**

Yes.

---

> ### Author Rebuttal · Authors · 2025-04-01
>
> Dear reviewer zysq:
>
> Thank you for your thoughtful suggestions and questions. We have provided point-by-point answers to each weakness and question.
>
> ## Weakness
> **W1. About theoretical innovation:**
>
> We acknowledge that our study may not introduce entirely new theoretical concepts, we believe that the theoretical support we provided is crucial for grounding our research in established frameworks, particularly in the context of invariant learning for predicting solvation properties.
>
> Our research contributes significantly to the field by demonstrating how invariant learning can effectively capture the complex relationships between molecular structures and their solvation properties. By integrating this methodology, we not only improve prediction accuracy but also provide a more interpretable framework for understanding the underlying molecular interactions.
>
> **W2. Figure 1 is intended to illustrate the invariance of interactions, but (a) the illustration is not easy to understand.**
>
> As shown in the figure, the horizontal axis represents solvent polarity, while the dotted box illustrates the distribution of different solute properties in the two solvents. Traditional methods predict solute properties by identifying core substructures; however, these core substructures may vary across different solvents. The key to our approach is to establish invariance by modeling interactions, which allows for a deeper understanding of the diverse characteristics of solute molecules in various solvents. The green arrows in the lower part of the figure indicate the interactions between molecules.
>
> **W3. There is no clear definition of what invariant interaction patterns are.**
>
> Invariant interaction patterns are features that remain consistent across various environments, or conditions. This involves identifying patterns that reflect interactions in diverse molecular contexts. Through invariant learning, we can extract key features from complex molecular data that capture fundamental interactions between molecules, independent of specific conditions. The model can still make accurate predictions even with unseen data or conditions. This capability is crucial for studying molecular interactions, as molecules may behave differently in various biological environments or experimental settings.
>
> **W4. Since obtaining explicit environment labels for solute-solvent pairs is often infeasible. So how is this method obtained?**
>
> We emphasize that environmental labels are not readily available, so we use auxiliary information to obtain environmental labels, i.e., other information that is easier to obtain. For example, the type of solvent, the scaffold of the molecule, etc., rather than the molecular interaction information obtained through complex modeling calculations as environmental labels.
>
> **Minor issues:** We’ve corrected errors and updated the description in the paper.
>
> ## Questions
> **Q1. Could you provide real-world examples to further illustrate the OOD generalization of RILOOD?**
>
> We provide solvent holdout case studies for your reference. Due to time constraints, we only selected a few cases. As can be seen from the case studies, the prediction error is still low for different types of solvents, including alkanes, alcohols, and olefins.
>
> Table. The results of RILOOD on solvents holdout test.
> |  Solvent  | RMSE |
> |-------|-------|
> | Tetraglyme  |  0.1004  |
> | Pentadecane |  0.1100  |
> | Tetradecane |  0.2462  |
> | 1-Dodecanol |  0.2315  |
> | 1-bromonaphthalene | 0.7796 |
> | Tetraglycol |  0.2015  |
>
>
> **Q2. About tsne, MCAR how to enhance feature diversity?**
>
> In fig.4(b), the feature distribution of MCAR is more dispersed than the original distribution, indicating that it contains more information and can better reflect the differences between samples after multi-interaction.
>
> We hope that our response will address your concerns and better clarify the contribution and value of our work.

---

### Official Review · Reviewer_Jwej · 2025-03-25

**Overall Recommendation:** 4

**Summary:**

This paper presents a novel out-of-distribution learning method for addressing the challenge of predicting solvation free energy in molecular interactions. The key innovation lies in the authors' approach to modeling the distribution of molecular interactions. They validated the effectiveness of their proposed model by testing it under out-of-distribution conditions.

**Claims And Evidence:**

A few statements lack adequate support, such as the one in line 277: “...the decisive interaction patterns of solute vary, leading to distinct solute properties.”

**Essential References Not Discussed:**

I note that MMGNN is similar to this work, but the authors does not make a comparison.

[1] MMGNN: A Molecular Merged Graph Neural Network for Explainable Solvation Free Energy Prediction

**Experimental Designs Or Analyses:**

The authors do not provide a specific description of the division of the dataset, such as the number of environments in the test set and the training set, which is not discussed in the experiment details and may make the comparisons unfair.

**Methods And Evaluation Criteria:**

The methodology and evaluation are reasonable.

**Other Comments Or Suggestions:**

- Fig. 1 uses some terms, such as polarity, dielectric constants, but they are not explained in the main text, what does it have to do with the invariance in this paper?
- $\mathbb{R} ^{N_x}$ in proposition 4.2, here, $N_x$ is missing definition.

**Other Strengths And Weaknesses:**

**Strengths**
- This paper addresses the OOD problem of molecular solvation free property prediction and is a pioneer work in this area.
- The proposed method relies on capturing the feature of molecular interactions that are invariant across environments, which is novelty.

**Weaknesses**
- The authors do not provide a specific description of the dataset partition settings, such as the number of environments in the test set and the training set, which are not discussed in the experimental setup and may make the comparisons unfair.
- This paper is similar to CMRL in that it also solves OOD problems through causality, please explain how it differs from CMRL?
- The hyperparameter alpha appears to be redundantly defined in both line 248 and Eq. 14. Additionally, the authors do not clarify how to choose alpha in line 248.
- Fig. 3 (c) is not detailed enough to provide a description of the specific tasks, please further elaborate the description.

**Questions For Authors:**

- Please explain “varying degrees between C and the label Y” in line 410?
- For other questions, please refer to the weakness.

**Relation To Broader Scientific Literature:**

This paper expands the scenarios of molecular applications and helps to expand to real-world scenarios.

**Theoretical Claims:**

Yes, I checked the theoretical claims.

---

> ### Author Rebuttal · Authors · 2025-03-31
>
> Dear reviewer Jwej:
>
> We sincerely appreciate the reviewer's thoughtful suggestions and questions. We have provided point-by-point answers to each weakness and question.
>
> ## Weakness
>
> **W1. Details about datasets:**
>
> We provided splitting details in appendix. The method of scaffold split is available in MoleculeNet, and the solvent split is the same as that of scaffold split. As shown in definition 1, we divide the dataset according to different environments, where scaffold and solvent are based on different environments.
>
> **W2. Different from CMRL:**
>
> CMRL acts as an invariant through the substructure of the interaction between molecular pairs, while at the same time weakens the influence of solvent molecules through backdoor adjustment. However, this invariance does not apply to solvent environment cases, because the interaction between solvents and solutes is multi-level, different levels of molecular interaction affect different properties. In contrast, our method recognizes that invariance is not consistent across different solvent environments due to the complex, multihierarchical nature of solute-solvent interactions, where varying interaction scales impact molecular properties differently.
>
> **W3. The choice of hyperparameter alpha:**
>
> For alpha, we chose 0.5 in order to get a uniform distribution.
>
> **W4. Minor issues:** Thanks for pointing out our omission. Figure 3 shows the results of the COMPSOLV dataset under 4 OOD settings. We've updated the graph annotations for fig.3 (c).
>
> ## Other Comments Or Suggestions
>
> **C1. Fig. 1 uses some terms, such as polarity, dielectric constants, but they are not explained in the main text, what does it have to do with the invariance in this paper?**
>
> Some of the proper nouns on the Fig. 1 is solvent related properties such as dielectric constant, polarity of solvent, which are included in solvent. By constructing multi-level interactions, the effects of these properties on solute molecules can be learned, and invariance can be further learned.
>
> **Differences from MMGNN:**
>
> MMGNN is a model for modeling the relationship between solvents and solutes, and our method focuses on the generalization of molecules in different solvents, which are fundamentally different. In contrast, our approach recognizes that invariance is not uniform in solvent environments due to the multihierarchical nature of solute-solvent interactions, where different interaction scales affect various molecular properties. To solve this problem, we introduce a multi-granularity context-aware refinement mechanism that captures local and global interactions to ensure robust performance under different solvent conditions.
>
> ## Claims And Evidence:
>
> The statement “the decisive interaction patterns of solute vary, leading to distinct solute properties” can be supported by references to PAR[1] and Unimatch[2]. We have included these references for further clarification.
>
> ## Questions
>
> **Q1: “...varying degrees between C and the label Y” in line 410?**
>
>  The spurious relationships of varying degrees between C and Y means that spurious correlations are introduced by controlling the distribution of variant subgraphs.
>
> We hope that our response will address your concerns and better clarify the contribution and value of our work.
>
> ## Reference
>
> [1] Property-Aware Relation Networks for Few-Shot Molecular Property Prediction
>
> [2] UniMatch: Universal Matching from Atom to Task for Few-Shot Drug Discovery

---

> > ### Comment · Reviewer_Jwej · 2025-04-07
> >
> > Thank you for the responses. The authors clarified the differences from previous works and addressed my concerns about the dataset partition settings. I also considered the comments and responses from other reviewers. Based on that, I would keep positive score.

---

> > > ### Author Response · Authors · 2025-04-08
> > >
> > > Thank you very much for your thoughtful comments. We are very grateful for your meticulous review and constructive criticism of our manuscript. Your comments are insightful and help us better understand how to improve our work.

---

### Decision · Program_Chairs · 2025-05-01

**Decision:**

Accept (spotlight poster)

**Comment:**

This paper proposed an approach for learning solvation free energy prediction of molecules, with a focus on OOD settings. The paper is well written and the empirical evaluation supports the claims in the paper. Reviewers have particular highlighted the OOD performance to be of pioneering value with a strong performance boost over existing work. The AC agrees that this paper is of value to the ICML community.